# Low Pathogenic Avian Influenza H9N2 Viruses in Morocco: Antigenic and Molecular Evolution from 2021 to 2023

**DOI:** 10.3390/v15122355

**Published:** 2023-11-30

**Authors:** Oumayma Arbani, Mariette F. Ducatez, Salma Mahmoudi, Faiçal Salamat, Slimane Khayi, Mohamed Mouahid, Karim M. Selim, Faouzi Kichou, Ikram Ouchhour, Mohammed El Houadfi, Siham Fellahi

**Affiliations:** 1Department of Veterinary Pathology and Public Health, Institut Agronomique et Vétérinaire Hassan II, Rabat 10000, Morocco; faicalsalamat651@gmail.com (F.S.); f.kichou@iav.ac.ma (F.K.); ouchhourikram@iav.ac.ma (I.O.); elhouadfimohammed@yahoo.fr (M.E.H.); 2Laboratoire Interactions Hôtes-Agents Pathogènes (IHAP), Toulouse University, INRAE, ENVT, 31300 Toulouse, France; mariette.ducatez@envt.fr; 3Laboratory of Microbiology and Molecular Biology, Department of Biology, Faculty of Sciences, Mohamed V University in Rabat, 4 Avenue Ibn Battouta, Rabat 10106, Morocco; salma.mahmoudi@um5r.ac.ma; 4Biotechnology Research Unit, CRRA-Rabat, National Institute of Agricultural Research, Rabat 10101, Morocco; slimane.khayi@inra.ma; 5Mouahid Veterinary Clinic, Temara 12000, Morocco; mohamedmouahid@gmail.com; 6Reference Laboratory for Veterinary Quality Control on Poultry Production, Agriculture Research Center, Animal Health Research Institute, Giza 12618, Egypt; dr.kareemseleem_87@yahoo.com

**Keywords:** H9N2, avian influenza, phylogenetic analysis, molecular characterization, Morocco, genetic diversity, zoonotic potential

## Abstract

Avian influenza viruses pose significant threats to both the poultry industry and public health worldwide. Among them, the H9N2 subtype has gained substantial attention due to its high prevalence, especially in Asia, the Middle East, and Africa; its ability to reassort with other influenza viruses; and its potential to infect humans. This study presents a comprehensive phylogenetic and molecular analysis of H9N2 avian influenza viruses circulating in Morocco from 2021 to 2023. Through an active epidemiological survey, a total of 1140 samples (trachea and lungs) and oropharyngeal swabs pooled into 283 pools, collected from 205 farms located in 7 regions of Morocco known for having a high density of poultry farms, were analyzed. Various poultry farms were investigated (159 broiler farms, 24 layer farms, 10 breeder farms, and 12 turkey breeder farms). A total of 21 AI H9N2 strains were isolated, and in order to understand the molecular evolution of the H9N2 avian influenza virus, their genetic sequences were determined using the Sanger sequencing technique. Phylogenetic analysis was performed using a dataset comprising global H9N2 sequences to determine the genetic relatedness and evolutionary dynamics of the Moroccan strains. The results revealed the continued circulation and diversification of H9N2 avian influenza viruses in Morocco during the study period. Real-time RT-PCR showed a positivity rate of 35.6% (73/205), with cycle threshold values ranging from 19.2 to 34.9. The phylogenetic analysis indicated that all Moroccan strains belonged to a G1-like lineage and regrouped into two distinct clusters. Our newly detected isolates aggregated distinctly from the genotypes previously isolated in Morocco, North and West Africa, and the Middle East. This indicats the potential of virus evolution resulting from both national circulation and cross-border transmission. A high genetic diversity at both nucleotide and amino-acid levels was observed among all the strains isolated in this study, as compared to H9N2 strains isolated in Morocco since 2016, which suggests the co-circulation of genetically diverse H9N2 variants. Newly discovered mutations were detected in hemagglutinin positions 226, 227, and 193 (H3 numbering), which highlights the genetic evolution of the H9N2 AIVs. These findings contribute to our understanding of the evolution and epidemiology of H9N2 in the region and provide valuable insights for the development of effective prevention and control strategies against this emerging avian influenza subtype.

## 1. Introduction

Avian influenza virus subtype H9N2 is a highly infectious pathogen that affects poultry and other bird species, causing significant economic losses in the poultry industry worldwide. It is classified as low pathogenic avian influenza virus (LPAIV), but its ability to reassort with other avian influenza viruses and potentially with human influenza viruses highlights its importance as a potential source of pandemic influenza virus genes. The virus has also been known to infect humans [1,2].

Earlier research has identified two separate lineages of H9N2 influenza viruses, namely the North American and Eurasian lineages. The Eurasian lineage can be subdivided into three main sub-lineages: the G1 lineage, represented by A/Quail/Hong Kong/G1/97 (G1-like); the Y280 lineage, which includes three prototype viruses, namely A/duck/Hong Kong/Y280/97 (Y280-like), A/Chicken/Beijing/1/94 (BJ94-like), and A/Chicken/Hong Kong/G9/97 (G9-like); and the Korean lineage, which encompasses A/chicken/Korea/38349-p96323/96 (Korean-like) and A/duck/Hong Kong/Y439/97 (Y439-like) [3,4,5].

H9N2 AIV was first isolated from turkeys in Wisconsin (USA) in 1966 (A/turkey/Wisconsin/1/1966(H9N2) [6]. By the early 1990s, the first cases of H9N2 influenza viruses were detected in poultry in China co-circulating with H5N1 [4,7]. Since then, it has become widespread in domestic chickens in numerous countries across Asia, the Middle East, and West and North Africa, where the virus has become endemic in poultry populations [8,9,10,11]. Meanwhile, virus circulation is the least limited in European countries in domestic as well as wild bird populations, as compared to highly pathogenic influenza (HPAI H5Nx), for which cases continue to be reported particularly in gull species [12]

The emergence of H9N2 viruses in Africa dates back to the early 2000s, with the first reported cases in Libya [13], followed by Egypt in 2006 [14] and Tunisia in 2009 [15]. In early 2016, the low pathogenic avian influenza (LPAI) H9N2 virus was identified in Morocco, resulting in significant economic losses for the poultry industry [16]. Since then, these viruses have spread significantly, not only throughout the country but also across North Africa, extending their reach to West and East Africa. Subsequently, Algeria detected H9N2 viruses in 2017 [17], and several Sub-Saharan African countries, including Burkina Faso [18], Ghana [19], Uganda, Kenya, Senegal [20], Benin, and Togo [21], have reported their presence as well. All H9N2 viruses detected in African poultry share a common G1 lineage and exhibit genetic relatedness with strains isolated in the United Arab Emirates. Previous studies have suggested a potential connection between the Moroccan isolate and the Middle East strain A/chicken/Dubai/D2506.A/2015 [22]. Despite the implementation of an extensive mass vaccination program for poultry, the virus has persisted and become endemic across all sectors of the poultry industry in Morocco [23].

The H9N2 virus has occasionally broadened its host range and gained mammalian host receptors. Therefore, the first isolated H9N2 virus in humans was in southern China in 1998 [24]. Studies on the mammalian adaptation of H9N2 viruses have revealed that most human isolates exhibit a leucine residue at position 226 (H3 numbering) of the hemagglutinin. This residue has been linked to a greater affinity for binding to α2,6 sialic acid, which is the most commonly found host cell receptor in the human upper respiratory tract. Recent human isolates have also been found to exhibit deletions in the stalk region of the NA protein, specifically at amino acid positions 62–64, which have been associated with mammalian adaptation. Further novel hemagglutinin (HA) mutations have been identified and associated with increased binding to human-like receptors [2,25,26]. Recent human isolates have also harbored deletions in the stalk region of the NA protein, specifically at amino acid positions 62–64, which have been associated with mammalian adaptation [1,25,27,28,29,30,31].

To date, the World Health Organization (WHO) has declared a total of 87 human cases infected with low pathogenic avian influenza (H9N2), including two deaths in the latest report published in April 2023 [12]. However, studies in humans exposed to poultry in endemic countries showed that many people harbor H9N2-specific antibodies, demonstrating that subclinical infections are common in several countries. However, no evidence of the human-to-human transmission of LPAI H9N2 viruses has yet been reported [32].

In the context of the critical spread of highly pathogenic influenza and the elevated risk of influenza virus reassortments, understanding the evolutionary history and circulation patterns of Moroccan H9N2 viruses becomes crucial for safeguarding public and animal health. Gaining deeper insights into the epidemiological and geographic origins of these viruses holds the potential to significantly enhance control and management strategies for future outbreaks. To address this pressing concern, our study aimed to comprehensively analyze the HA sequences of H9N2 avian influenza viruses collected in Morocco from 2021 to 2023. By monitoring the relevant site of mutations, we sought to provide valuable knowledge for proactive measures against the spread of H9N2.

## 2. Materials and Methods

### 2.1. Sample Collection

From our epidemiological surveys launched from September 2021 to March 2023 in seven regions of Morocco. We collected 1140 samples, including trachea and lungs, and oropharyngeal swabs, which were pooled into 282 pools collected from 205 farms (vaccinated and non-vaccinated) of different types of production (159 broiler, 24 layer,10 breeder, and 12 turkey breeder). The samples were at least from 10 samples per farm and were pooled subsequently into 1 or 2 pools for RT-PCR analysis. This pooling approach was designed to ensure that the farm’s status (whether positive or negative) was represented adequately and accurately.

Our sampling was based on flocks suspected of being infected with the LPAI H9N2 virus and presenting respiratory/ocular signs including sneezing, coughing, rales, excessive lacrimation, and rattles, which are associated with a decrease in food consumption and a drop in production and sometimes with high rates of morbidity and mortalities.

Samples of tissues were collected from freshly dead birds based on macroscopic respiratory signs related to the virus (fibrinous casts in tracheal bifurcation, congestive and/or fibrinous tracheitis, pneumonia, and sinusitis). Tracheal swabs were collected from live birds and prepared as described by the WOAH (World Organization for Animal Health).

### 2.2. Virus Extraction and Real-Time RT-PCR for H9 Detection

Viral RNA was extracted from pools of oropharyngeal swabs, homogenized tissues, or infective allantoid fluid, using a Kylt^®^ RNA/DNA Purification Kit according to the manufacturer’s directions (AniCon Labor GmbH Laboratory, Emstek, Germany). Samples were first analyzed by real-time RT-PCR using H9-specific primers and probe sets for conserved regions in the HA2 subunit, as described by [33], using an Invitrogen™ SuperScript™ III One-Step RT-PCR Kit (Thermo Fisher Scientific, Göteborg, Sweden) on a 7500 Real-Time PCR System (Applied Biosystems, Foster City, CA, USA).

### 2.3. Virus Isolation and cDNA Purification

Highly positive samples with qRT-PCR (CT < 30) were subjected to virus isolation on 9- to 11-day-old specific pathogen-free chicken embryonated eggs. Inoculated embryos were incubated at 37 °C for 72 h and then refrigerated at 4 °C for 4 h before harvesting. Harvested allantoid fluid was clarified by centrifugation and then stored at −20 °C until use.

The isolates obtained were subjected to conventional RT-PCR for sequencing. In order to amplify the HA gene using DNA polymerase, the extracted genetic material (RNA) was first converted to complementary DNA (cDNA) by reverse transcription (RT) using an Applied Biosystems Kit (Life Technologies) before a PCR amplification was performed using a Dream TAQ Green PCR MM(×2) Kit (Thermo Fisher Scientific). The primers described by Hofmann et al. (2001) [34] were used for HA gene amplification with a predicted size of the product of approximately 670 bp on a 1% agarose gel. All amplicons corresponding to this size were excised and purified from the gel using a Machery–Nagel kit (Nucleospin gel and PCR clean-up), following the manufacturer’s instructions.

### 2.4. Sequencing and Phylogenetic Analysis

The purified products were sent for Sanger sequencing to Eurofins (Hamburg, Germany). In this study, we conducted a sequence analysis on a total of 21 HA gene segment of the H9N2 virus, focusing on variable regions within the range of 180 to 250 amino acids. For 12 of these segments, we performed Sanger sequencing, covering the HA cleavage site and the variable segment of both HA1 and HA2 subunits (around positions NT 57 to 328 on HA1 and 1 to 176 for HA2 (H3 numbering)). In the case of the remaining nine segments, our sequencing method was directed solely towards the HA2 subunit (around positions NT 1 to 223 (H3 numbering)).

The raw sequences obtained in AB1 format were aligned and analyzed using Bioedit 7.2.5 Software [35]. Using the maximum composite likelihood method, a phylogenetic tree was constructed with MEGA 6.06 software following the Tamura model [36]. The robustness of the groupings in the maximum likelihood analysis was assessed with 1000 bootstrap replicates. The sequences obtained were compared with the H9N2 genomic sequences available in the GenBank database using Nucleotide BLAST provided by NCBI [37] and ClustalW on Bioedit 7.2.5 Software [38]. All the nucleotide strains characterized in this study were submitted to GenBank and were attributed accession numbers, which are presented in Table 1.

## 3. Results

### 3.1. Epidemiological Survey and Real-Time RT-PCR H9N2 Results

Real-time RT-PCR showed that 73 farms out of 205 tested positive, resulting in a positivity rate of 35.6%. Overall, 36.4% (58/159) of broiler farms, 16.6% (4/24) of layer farms, 50% (5/10) of chicken breeder farms, and 50% (6/12) of turkey breeder farms were H9-positive, respectively. Cycle threshold (Ct) values ranged from 9.2 to 34.9, with 33%, 61%, and 5% of the samples having a Ct below or equal to 25, between 25 and 35, and above 35, respectively.

The distribution of the positive samples in the farms tested is presented per geographical region (Figure 1).

Figure 2 concludes with the sum totals, presenting the overall counts of farms based on their representative statues (negative or positive) across the entire dataset. These statuses are determined based on positive samples in RT-PCR. The data are divided by both region and type of farm, offering a glimpse into farm statuses throughout various regions of Morocco and based on different production types.

On the basis of the Ct, the geographical area, and the type of production, a total of 21 representative positive samples were selected for partial genome sequencing.

### 3.2. Phylogenetic Analysis of HA Gene Segment

Blast analysis of the nucleotide sequences from the 21 viral genes showed that our isolates were closely related to the previous Moroccan H9N2 strains isolated since 2016, with a nucleotide identity ranging from 94.3% to 98.2%. These isolates also had nucleotide identities of 95.4 to 98.0% with strains from North Africa, especially those from Algeria and the environment in Tunisia, as well as nucleotide identities ranging from 95.1% to 97.2% with West African strains, such as those identified in Benin and Nigeria in 2019. The percent of nucleotide identities in comparison with the Moroccan isolates from the present study and representative groups of subtypes H9N2 are shown in Table A1.

The complete analysis of the haemagglutinin (HA) gene segment of these viruses collected between 2021 and 2023 was performed using maximum likelihood phylogenetic analysis, which unveiled intriguing findings regarding the Moroccan H9N2 subtype viruses. The ML phylogenetic analysis revealed that LPAIV H9N2 Moroccan viruses belong to the G1 lineage and regroup to two main distinct clusters distinguished from the previously isolated genotypes.

The first cluster (*n* = 13) includes the new Moroccan viruses collected dominantly from broilers, one layer farm, and Turkey breeders located in the North, Northwest, and East Regions of Morocco, around the regions of Rabat, Casablanca, Marrakech, Fes, and Oujda. In the same branch, viruses isolated in Morocco in 2016, 2017, 2018, and 2020, as well as viruses from North Africa, Algeria (2017), and Tunisia (2018), are clustered.

The second group (*n* = 8) consists of Moroccan strains isolated from three layer farms and broilers located in the Northwest region, in the Rabat and Casablanca regions. These viruses cluster closely with LPAIV H9N2 viruses isolated in Morocco in 2020 but less with viruses isolated in Benin (2019, 2020); layers in Senegal (2017), Nigeria (2019), and Togo (2019); and previous strains from Morocco isolated between 2016 and 2020. These findings are depicted in Figure 3.

In Figure A1, the analysis of the HA median-joining network confirms the evolution of the virus into different genetic clusters. The divergence of the strains isolated recently from the ancestors, especially strains isolated in 2016 and 2017, is remarkable. Our strains isolated from 2021 to 2023 diverge from the longest branches from the center in different directions, exhibiting high nucleotide differences and forming distinct clusters.

### 3.3. Genetic Diversity

This analysis was conducted to assess the genetic diversity among H9N2 strains isolated during our study in Morocco, as well as previous H9N2 viruses from Morocco, Algeria, and Tunisia, and a reference H9N2 virus from Hong Kong. These investigations involved calculating evolutionary distances between different genotypes (Table 2).

The results revealed a significant genetic diversity within the strains grouped in the two main clusters and the reference H9N2 virus from Hong Kong isolated in 1997. The genetic diversity ranged from 8% to 10% for nucleotide sequences and from 3% to 6% for amino acid sequences within these groups. This indicates that while some H9N2 strains are closely related in both amino acid and nucleotide sequences, Hong-Kong/G1/1997 appears to be genetically distinct from the others.

When comparing our H9N2 strains with those showing the highest genetic similarities (Moroccan, Algerian, and Tunisian viruses), the new isolates exhibit moderate nucleotide and amino-acid genetic distances from other strains. The genetic distance ranges from 2% to 6% for nucleotide sequences and from 1% to 6% for amino acid sequences, suggesting a relatively divergent genetic relationship.

### 3.4. Molecular Analysis

Amino acid sequences of the HA gene from our H9N2 viruses show significant variations observed at several positions when compared to previously isolated strains. The sequencing of the variable HA1 and HA2 regions was conducted for 12 strains out of the 21 sequenced. In the strains where only HA2 was sequenced, only potential glycosylation sites were investigated, as we did not have enough data to present the mutations on RBS.

All those isolated had the RSSR*GLF motif at the HA cleavage site, which is a characteristic of the low pathogenic H9N2 viruses.

Several mutations were noted among our H9N2 viruses, mainly at position 226 (H3 numbering), where presenting serine (S) instead of leucine (L) or arginine (A) was previously noted. Additionally, glutamine (Q) was the most frequent determinant at position 193. Furthermore, the pattern of observed mutations at positions 191, 193, 195, 196, 197, 228, and 230 compared to strains isolated in Morocco in 2016 and 2018, were dominantly Q191G, N193Q, Y195K, T196R, R197Q, G228R, and I230R, which are all summarized in Table 3.

However, no mutation was noted at position 194, which conserved the amino-acid leucine (L).

For the potential HA glycosylation sites, our sequencing did not include the entire HA gene. The main glycosylation sites were identical to previous Moroccan viruses and noted at the HA2 subunit at positions 154 NGTY and 213 NGSC. Four out of twenty-one isolates had one additional glycosylation site at position 95 NASC.

## 4. Discussion

The present study was conducted from September 2021 to March 2023 across seven regions of Morocco. The objective of the study was to isolate and characterize (at the molecular level) the strains circulating in poultry farms within the country. A total of 1140 samples pooled into 282 pools were collected from 205 farms, encompassing various types of poultry production. These samples were suspected to be infected with the low pathogenic avian influenza (LPAI) virus subtype, H9N2. The results of this phylogenetic and molecular analysis of H9N2 avian influenza viruses in Morocco provided valuable insights into the genetic diversity and evolutionary dynamics associated with this subtype. Starting with molecular analysis, real-time RT-PCR showed a positivity rate of 35.6%, with the virus being present in at least six of the seven regions of Morocco tested. The viropositivity rate observed in our survey is notably lower than the previously reported in 2019, which was 58% in 108 farms, as reported by a previous study [39]. It is important to note that both studies included samples collected from suspected farms, presenting high mortality rates, respiratory signs, a decrease in food consumption, and a drop-in production. This decline in viropositivity rate could suggest several possibilities: changes in the prevalence of the virus, with a co-circulation of multiple respiratory pathogens with the H9N2 virus, especially infectious bronchitis virus [40], which causes the same respiratory clinical picture; improved biosecurity measures; or fluctuations in disease dynamics within the poultry industry. Nonetheless, the vaccination programs implemented in poultry production sectors using local and non-local strains since the first introduction of the virus in Morocco in 2016 [16,41] may play an important role in this decrease. Many studies revealed that adapted vaccination reduces the level and duration of virus shedding and increases the resistance to host infection [8,42]. Moreover, post-vaccination samples were found to indicate a limited transmission of the virus in the field [32,43]. On the other hand, vaccination programs may result in faster antigenic drift of human and avian influenza viruses [32,44]. However, vaccination pressure and natural immune pressure due to endemic infections in unvaccinated flocks contribute to the evolution and genetic variation of HA gene segments [45]. Therefore, in the context of endemic LPAIV H9N2 in several African countries, especially in the neighboring North Africa region, it is essential to extend vaccination programs to cover the entirety of poultry production. This approach of vaccination in breeders, layers, and turkey breeders has been launched in Algeria, Tunisia, and Egypt but is less extended in broilers [10,17,46]. Thus, updating vaccine strains based on ongoing surveillance surveys and circulating genotypes is a crucial factor for the epidemiological control of virus spreading and for limiting H9N2 circulation.

The focus on farms experiencing suspected health issues remains crucial as it allows for a deeper investigation into the specific challenges faced by these farms and can potentially aid in the development of more targeted interventions and control strategies. In addition, most of the investigated farms in our study were predominantly situated in Northwest regions (Rabat, Casablanca, and nearby areas). This particularity was a direct result of the heightened poultry activity and growing industry in this geographical region. We acknowledge that the sampling distribution may not provide a diverse description of the entire national territory. However, this focus allows us to gain insight into the areas where the virus is spreading most frequently.

The phylogenic analysis showed that all Moroccan H9N2 viruses isolated since 2016 belong to the G1 lineage. Homology analysis revealed the presence of multiple distinct genotypes related to previously isolated strains with nucleotide similarity ranged between 94.31% to 98.20%. Two distinct clusters were reported circulating in the national territory. Cross-transmission occurred between two clusters were noted in the Northwest region, where livestock is most concentrated. The newly detected viruses in our study gathered distinctly from the old ones. However, they kept the genetic relatedness with strains isolated from Morocco (2016, 2018, 2019 and 2020), Algeria (2017), Tunisia (2018), Benin (2019,2020), Senegal (2017), Nigeria (2019) and Togo (2019). This closeness is suggestive of shared genetic ancestry or a recent common origin [16,41,47,48]. As proximity facilitates the exchange between these neighboring countries, the frequent trade and movement can serve as a vector of the transmission of influenza viruses. This interaction can lead to the sharing of influenza strains and the emergence of genetically related viruses.

In addition, H9N2 AI virus circulating in Morocco exhibited considerable variation in nucleotide and amino acid motifs, in comparison with strains isolated since 2016, with a genetic diversity ranging from 2% to 6% for nucleotide sequences and from 1% to 6% for amino acid sequences, which separates our 2021, 2022 and 2023 from their ancestors. With a high vaccination rate and the use of many vaccines, this shows continued genetic diversity and evolutionary changes within the virus population. These isolates’ virulence, transmissibility, and capacity for interspecies transmission may be affected by the genetic diversity among them [31]. The primary driving factors in the evolution of AIV viruses are antigenic drifts and transformation. Nucleotide substitution rate is considered as one of the main index to assess this evolution [49]. Nonetheless, many factors can contribute to heightened nucleotide substitution rate, such as suboptimal vaccination procedures, competition among distinct branches within the same AIV subtype, genetic reassortment and gene recombination [44,50,51]. Receptor binding site motif of the HA protein has been extensively studied and is crucial for the binding of influenza viruses to host epithelial cells and subsequent fusion with host endosomal membranes [31,52,53]. The receptor binding specificity of influenza viruses is strongly influenced by specific amino acid residues, particularly at positions 226 and 228 (H3 numbering). A leucine (L) residue at position 226 and a serine (S) residue at position 228 promote binding to receptors similar to those found in humans. Conversely, a glutamine (Q) residue at position 226 and a glycine (G) residue at position 228 enhance binding to receptors typically found in avian species [54]. Another important substitution occurs at residue 193 of the HA protein, specifically N193E, which has been linked to preferences for both human and avian-like receptors. Notably, viruses carrying the D, E, or S substitutions at residue 193 have demonstrated the ability to replicate in the lungs of infected chickens, while the G193 variant has shown reduced transmissibility among chickens in direct contact [55]. One well known HA substitution, Q226L, was detected in older Moroccan strains [16,39,41], significantly increases the binding of A(H9) AIVs to α2,6-linked sialic acids, and enhances AIV transmission in ferrets [53,56]. Our study has revealed newly detected mutations in crucial RBS positions; L226S, G228R and E193T. Which can harbor mammalian or other hosts adaptive variations, promote preferential binding to sialic acid receptors of epithelial cells and escape to immune mechanisms [29,57]. The variations at HA residue 193 have been found to affect various properties of H9N2 viruses, underscoring the importance of continuous surveillance of the HA protein for a better understanding of H9N2 etiology and effective control in poultry [55]. Other substitutions identified in HA segment of A(H9N2) viruses have also been detected which can be associated with AIV replication in mammals and increased virulence. Further work is needed to see if increased zoonotic transmission, virulence, and resistance to anti-viral has occurred with this change in receptor specificity. We have also noted the deletion of two potential glycosylation sites, which were proposed to influence the virulence and the adaptation of AIV to poultry [57,58]. Furthermore, differences in glycosylation of the virion surface proteins may contribute to antigenic variations, however this requires further evaluation [59]. In summary, the A(H9N2) avian influenza viruses isolated from poultry farms between 2021 and 2023 were identified as belonging to the G1 lineage. These viruses showed close genetic connections to strains found in neighboring countries, indicating the possibility of frequent circulation due to the movement of live poultry across borders. Additionally, wild birds might also contribute to the transmission of A(H9) in the region, but surveillance efforts in wild birds are too scarce in the region to allow us to verify this hypothesis. Furthermore, newly detected Moroccan A(H9N2) viruses harbor new molecular markers that may increase their potential adaptation to mammalian species, raising concerns about their potential to infect humans.

Overall, the ongoing presence and evolution of A(H9N2) avian influenza viruses in Moroccan poultry emphasizes the importance of continuous and vigilant surveillance and analysis of avian influenza viruses within the country to monitor and manage any potential risks to both poultry and public health.

## 5. Conclusions

The results discussed in our study suggest ongoing genetic diversity and evolutionary changes within the viral population of H9N2 in the national territory. The co-circulation of diverse strains indicates the potential for antigenic drift, which impacts the effectiveness of existing vaccines and diagnostic tools, presenting the necessity of adapting vaccines and programs to the circulating strains and the epidemiological situation of each region. Moreover, the close genetic relatedness of Moroccan strains to those from neighboring countries suggests cross-border transmission, emphasizing the need for regional surveillance and coordinated efforts to control the spread of the virus.

## Figures and Tables

**Figure 1 viruses-15-02355-f001:**
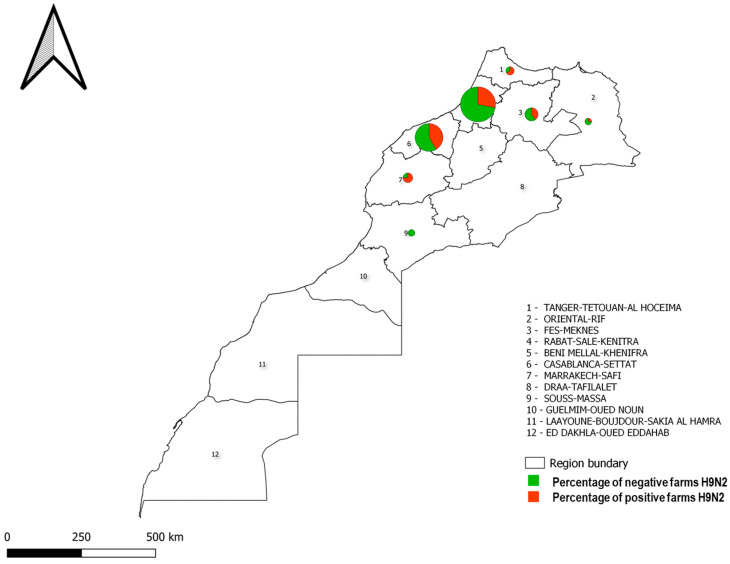
Geographical distribution of H9N2 AIV strains in Morocco between 2021 and 2023.

**Figure 2 viruses-15-02355-f002:**
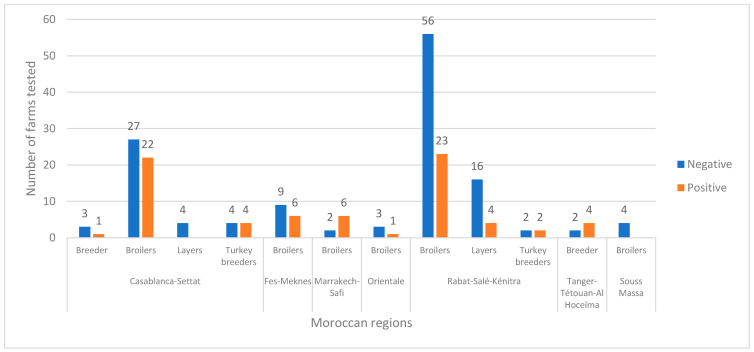
Distribution of tested farms in seven Moroccan regions.

**Figure 3 viruses-15-02355-f003:**
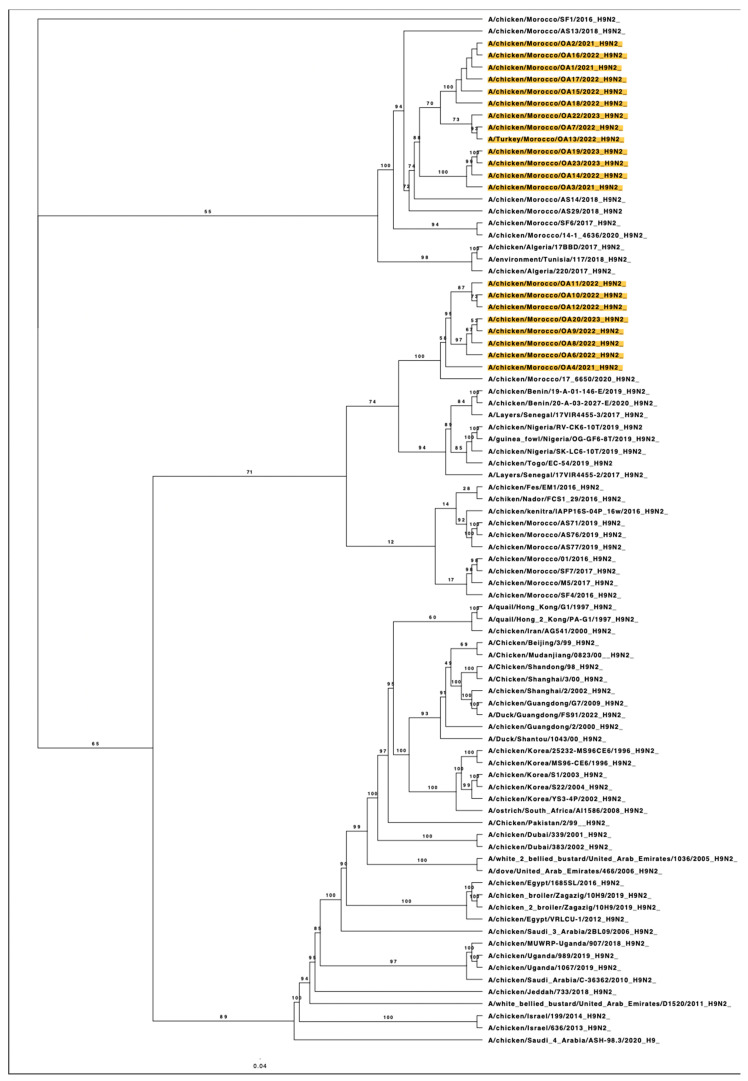
Maximum-likelihood phylogenetic tree of the HA gene segments of representative LPAI H9N2 viruses. Moroccan viruses sequenced for the purpose of this study are marked in yellow.

**Table 1 viruses-15-02355-t001:** Related information of studied Moroccan H9N2 viruses and NCBI accession numbers.

Virus Identification	Date	Location	Type of Production	Age of Birds *	Tissue	Ct Value of qRT-PCR	Accession Number
A/chicken/Morocco/OA1/2021	07/04/2021	Rabat	Layers	34w	Trachea and lung	29.09	OR592450
A/chicken/Morocco/OA2/2021	02/08/2021	Casablanca	Broilers	32d	Trachea and lung	23.69	OR592441
A/chicken/Morocco/OA3/2021	01/12/2021	Rabat	Broilers	34d	Oropharyngeal swabs	10.29	OR592474
A/chicken/Morocco/OA4/2021	17/12/2021	Temara	Broilers	19d	Oropharyngeal swabs	14.71	OR592443
A/chicken/Morocco/OA6/2022	27/01/2022	Kénitra	Broilers	22d	Oropharyngeal swabs	10.84	OR592455
A/chicken/Morocco/OA7/2022	31/01/2022	Kénitra	Broilers	18d	Trachea and lung	12.02	OR592451
A/chicken/Morocco/OA8/2022	14/03/2022	Salé	Broilers	31d	Oropharyngeal swabs	13.05	OR592475
A/chicken/Morocco/OA9/2022	14/03/2022	Salé	Broilers	31d	Trachea and lung	11.56	OR592452
A/chicken/Morocco/OA10/2022	14/11/2022	Rabat	Layers	34w	Trachea and lung	26.07	OR592453
A/chicken/Morocco/OA11/2022	22/11/2022	Rabat	Layers	25w	Trachea and lung	28.08	OR592442
A/chicken/Morocco/OA12/2022	29/01/2022	Casablanca	Broilers	26d	Trachea and lung	25.22	OR592454
A/Turkey/Morocco/OA13/2022	19/11/2022	Bouznika	Turkey breeders	42d	Trachea and lung	25.06	OR592514
A/chicken/Morocco/OA14/2022	15/02/2022	Marrakech	Broilers	24d	Trachea and lung	25.75	OR592449
A/chicken/Morocco/OA15/2022	10/12/2022	East region	Broilers	35d	Trachea and lung	25.25	OR592456
A/chicken/Morocco/OA16/2022	10/12/2022	East region	Broilers	41d	Trachea and lung	24.85	OR592448
A/chicken/Morocco/OA17/2022	07/11/2022	Rabat	Broilers	19d	Trachea and lung	23.72	OR592457
A/chicken/Morocco/OA18/2022	09/11/2022	Fes	Broilers	27d	Trachea and lung	21.92	OR592447
A/chicken/Morocco/OA19/2023	03/01/2023	Meknes	Broilers	21d	Trachea and lung	19.44	OR592444
A/chicken/Morocco/OA20/2023	13/01/2023	Casablanca	Layers	30w	Trachea and lung	28.6	OR592513
A/chicken/Morocco/OA22/2023	12/01/2023	Fes	Broilers	47d	Trachea and lung	23.89	OR592446
A/chicken/Morocco/OA23/2023	01/03/2023	Temara	Broilers	28d	Trachea and lung	28.68	OR592445

* W = weeks, d = days.

**Table 2 viruses-15-02355-t002:** Evolutionary genetic distance for nucleotide and amino-acid sequences between the four clusters of the Moroccan strains.

	1	2	3	4	5	6	7	8	9	10	11	12	13	14	15	16
	Amino-Acid Genetic Distance															
SF1/2016		0.004	0.09	0.004	0.009	0	0.002	0.004	0.025	0.02	0.016	0.03	0.011	0.032	0.032	0.011
AS13/2018	0.01		0.092	0.007	0.013	0.005	0.005	0.007	0.03	0.015	0.011	0.025	0.017	0.04	0.027	0.011
Hong_Kong/G1/1997	0.117	0.128		0.09	0.094	0.033	0.092	0.095	0.05	0.051	0.043	0.06	0.039	0.048	0.06	0.039
Algeria/220/2017	0.007	0.02	0.123		0.009	0	0.005	0.007	0.025	0.02	0.016	0.03	0.011	0.032	0.032	0.011
Tunisia/117/2018	0.016	0.03	0.124	0.019		0	0.011	0.013	0.025	0.02	0.016	0.03	0.011	0.032	0.032	0.011
Fes/EM1/2016	0	0.018	0.071	0.004	0		0	0	0.011	0.011	0.016	0.016	0.011	0.033	0.033	0.011
SF7/2017	0.002	0.015	0.119	0.008	0.018	0.002		0.005	0.025	0.02	0.016	0.03	0.011	0.032	0.032	0.011
Morocco/14-1_4636/2020	0.019	0.022	0.132	0.026	0.034	0.016	0.021		0.026	0.021	0.016	0.031	0.011	0.033	0.033	0.011
OA2/2021	0.04	0.045	0.101	0.043	0.04	0.027	0.041	0.047		0.025	0.027	0.035	0.022	0.032	0.043	0.022
OA13/2022	0.035	0.027	0.099	0.039	0.035	0.024	0.034	0.031	0.044		0.016	0.02	0.022	0.04	0.033	0.022
OA23/2023	0.036	0.025	0.087	0.039	0.036	0.037	0.038	0.03	0.041	0.038		0.021	0.028	0.04	0.038	0.022
OA3/2021	0.052	0.039	0.102	0.055	0.051	0.035	0.053	0.049	0.052	0.044	0.016		0.028	0.056	0.038	0.022
OA11/2022	0.026	0.044	0.088	0.03	0.026	0.026	0.028	0.038	0.043	0.05	0.048	0.05		0.024	0.022	0
OA20/2023	0.048	0.069	0.092	0.051	0.048	0.046	0.051	0.063	0.06	0.074	0.063	0.071	0.013		0.04	0.025
OA6/2022	0.034	0.047	0.093	0.038	0.034	0.031	0.036	0.046	0.052	0.054	0.051	0.052	0.009	0.018		0.016
OA4/2021	0.026	0.041	0.086	0.029	0.026	0.026	0.027	0.037	0.048	0.048	0.048	0.05	0.011	0.025	0.014	
	Nucleotide Genetic Distance															

**Table 3 viruses-15-02355-t003:** HA mutation on molecular determinants of receptor binding preferences in Moroccan H9N2 AIVs.

Strains	Position NT (H3 Numbering)				Receptor Binding Sites (H3 Numbering)									
	196	197	198	230	191	192	193	194	195	221	225	226	227	228
Beijing/3/99 ^a^	N	R	R	I	N	N	K	C	D	*	V	A	R	S
Hong_2_Kong/1997 ^b^	I	R	N	I	Q	T	N	L	Y	P	G	L	Q	G
SF1/2016 ^c^	T	R	T	I	Q	T	N	L	Y	P	G	L	I	G
AS14/2018 ^d^	T	R	T	I	Q	T	N	L	Y	P	G	L	I	G
AS77/2019 ^e^	T	R	T	I	Q	T	N	L	Y	P	G	L	I	G
17-6650/2020 ^f^	T	R	T	I	Q	T	N	L	Y	P	G	L	I	G
OA1/2021 ^g^	R	Q	L	R	G	F	Q	L	K	M	N	S	V	R
OA2/2021 ^h^	R	Q	L	R	G	F	Q	L	K	M	N	S	V	R
OA15/2022 ^i^	R	Q	L	R	G	F	Q	L	K	M	N	S	V	R
OA16/2022 ^j^	R	Q	L	R	G	F	Q	L	K	M	N	S	V	R
OA17/2022 ^k^	R	Q	L	R	G	F	Q	L	K	M	N	S	V	R
OA18/2022 ^l^	R	Q	L	R	G	F	Q	L	K	M	N	S	V	R

^a^ = A/Chicken/Beijing/3/99(H9N2); ^b^ = A/quail/Hong_2_Kong/PA-G1/1997(H9N2); ^c^ = A/chicken/Morocco/SF1/2016(H9N2; ^d^ = A/chicken/Morocco/AS14/2018(H9N2); ^e^ = A/chicken/Morocco/AS77/2019(H9N2); ^f^ = A/chicken/Morocco/17_6650_21RS1333-19/2020|A_/_H9N2; ^g^ = A/chicken/Morocco/OA1/2021(H9N2); ^h^ = A/chicken/Morocco/OA2/2021(H9N2) ^i^ = A/chicken/Morocco/OA15/2022(H9N2); ^j^ = A/chicken/Morocco/OA16/2022(H9N2); ^k^ = A/chicken/Morocco/OA17/2022(H9N2); ^l^ = A/chicken/Morocco/OA18/2022(H9N2).

## Data Availability

Data are contained within the article.

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
