# Peer review of "Low Pathogenic Avian Influenza H9N2 Viruses in Morocco: Antigenic and Molecular Evolution from 2021 to 2023"

_viruses, 2023, doi:10.3390/v15122355_

Round 1

Reviewer 1 Report

Comments and Suggestions for Authors

The paper by Arbani O. et al. presents a study on avian H9N2 influenza in Morocco. This study is of high value as H9N2 viruses often donate their internal genes to other influenza viruses, increasing the probability of generating highly pathogenic viruses.

Overall, the article leaves a good impression. The Introduction is well-written, clear, and provides all relevant data to introduce the reader to the problem described further. The materials and methods section is also clearly written, but the results section requires some clarification in order to properly assess the prevalence of the H9N2 virus in Morocco and the region. Additionally, the sequencing results seem incomplete as only fragments of the HA for isolated viruses were sequenced, which is surprising. However, this may be due to limited resources. The analysis of the data presented and the discussion section are satisfactory.

The general impression is good, and the article may be published after addressing the following issues:

Introduction: Lines 25-27 lack a predicate. Please correct this.

Line 84: Possibly "the" is missing before H9N2.

Materials and methods: In the sequencing section, please clearly indicate the length of the HA fragment that you were able to sequence and the positions of amino acids. Also, provide clarification on how many HAs were sequenced and whether they were all of the same length.

Results:

The main question is: In section 3.1, you indicate that you examined 205 farms. How many farms are there in Morocco? Is 205 the total number or a given percentage? Please indicate this. Moreover, you collected 282 samples from 205 farms, which means less than 2 samples per farm. In that case, it is not clear whether the further results are representative because 1-2 samples per farm is a very low sampling rate.

Additionally, the percentage in Figure 1 is not clear without further details. If one sample collected at the farm was enough to determine whether it was positive or negative, then the result is poor. Please provide more details. Also, correct the caption for Figure 1 from French to English (change "pourcentage" to "percentage"). The same applies to Figure 2. Does it show the number of farms or the number of samples?

For section 3.2, it would be helpful to know the exact amino acid length of the analyzed HA fragments and whether they were sequenced from the start. Does it include the full-length HA1 and partial HA2? Table 2 should be moved to supplementary material as it is very technical and already well-described in the text. Figures 3 and 4 are very similar, so please select only one to include in the main text, and move the other to supplementary material.

In section 3.3, line 244 states that six strains were sequenced for HA1 and HA2. Please provide more details. Were these full-HA sequences? And were the others not sequenced? Line 253 seems to be a fragment, so please rephrase it. In Table 4, the footnote for a and b should be reversed.

Comments on the Quality of English Language

minor editing is needed

Author Response

Answer to the Reviewer 1:

We extend our gratitude to the Reviewer for their valuable comments and for affording us the opportunity to enhance our manuscript for resubmission to Viruses. We have carefully considered the comments provided and have made the necessary revisions to the manuscript. Below, we provide a detailed, point-by-point response to the questions and comments raised by the Reviewer.

Reviewer 1

Comment 1: Introduction: Lines 25-27 lack a predicate. Please correct this.

Thank you for pointing this out. We have corrected it now in the manuscript.

Comments 2: Line 84: Possibly "the" is missing before H9N2.

We agree with the Reviewer and now we have corrected it. Please see the manuscript.

Comments 3: Materials and methods: In the sequencing section, please clearly indicate the length of the HA fragment that you were able to sequence and the positions of amino acids. Also, provide clarification on how many HAs were sequenced and whether they were all of the same length.

We agree with the Reviewer and we have now clarified this important point in the text.

It reads in 2.4 section (lines 152) as follows:

“The purified products were sent for Sanger sequencing to Eurofins (Germany). In this study, we conducted sequence analysis on a total of twenty-one HA gene segment of H9N2 virus, focusing on variable regions within the range of 180 to 250 amino acids. For 12 of these segments, we performed Sanger sequencing, covering the HA cleavage site and the variable segment of both HA1 and HA2 subunits (around position NT 57 to 328 on HA1 and 1 to 176 for HA2 (H3 numbering)). In the case of the remaining nine segments, the same sequencing method was directed solely towards the HA2 subunit (around position NT 1 to 223 (H3 numbering).”

Comments 4: Results: The main question is: In section 3.1, you indicate that you examined 205 farms. How many farms are there in Morocco? Is 205 the total number or a given percentage? Please indicate this. Moreover, you collected 282 samples from 205 farms, which means less than 2 samples per farm. In that case, it is not clear whether the further results are representative because 1-2 samples per farm is a very low sampling rate.

We thank the reviewer for your kind comments. The Reviewer is correct.

Indeed, Morocco has a significantly larger number of farms(more than 10 000 farms for all type of productions; however, broilers has the higher part with about 7000 farms). For our study, our sampling corresponds to 1140 organs and tracheal swab which were pooled to 282 poolcollected from 205 farms. It’s may seem low but it was interested only the suspected flocks infected with low pathogenic avian influenza H9N2 showing severe respiratory signs including sneezing, coughing, rales and gasping, associated with sinusitis, high mortality, decrease in feed consumption and a drop in production.The specific aim of our study is to present the genetic diversity and the molecular mutations newly detected on the wild and virulent strains circulating in Morocco this two recent years, so our target is the farms with symptoms related to LPAI H9N2.

On the other hand, the samples were between five and ten samples per farm and pooled subsequently to one or three pools for RT-PCR analysis.

We have now clarified this important point in the materiel and methods section 2-1 (line 115). It reads as follows:

“From our epidemiological surveys launched from September 2021 to March 2023 in seven regions of Morocco, we collected 1140 samples including (trachea and lungs) and oropharyngeal swabs which were pooled to 282 pool collected from205 farms (vaccinated and non-vaccinated) of different types of production (159 broiler, 24layer,10 breeder, and 12 turkey breeder). The samples were at least from 10 samples per farm and pooled subsequently to one or two pools for RT-PCR analysis. This pooling approach was designed to ensure that the farm's status (whether it is positive or negative) was represented adequately and accurately.”

Comment 5: Additionally, the percentage in Figure 1 is not clear without further details. If one sample collected at the farm was enough to determine whether it was positive or negative, then the result is poor. Please provide more details. Also, correct the caption for Figure 1 from French to English (change "pourcentage" to "percentage"). The same applies to Figure 2. Does it show the number of farms or the number of samples?

We agree with the Reviewer, the percentages and the caption were rectified in figure1.

As we previously explained, the sampling was representative in each farm tested so the positivity rate was determined based on the positive samples of each farm.

In figure 2 shows the number of farms in different regions with their "Negative" and "Positive" status for each type of production.

To further clarify we added a paragraph in Results, section 3-1 (line 171):

“The Figure 2 concludes with the sum totals, presenting the overall counts of farms based on their representative statues (Negative or Positive) across entire dataset. These statuses are determined based on positive samples on RT-PCR. The data is divided by both region and type of farm, offering a glimpse into farm statuses throughout various regions of Morocco and by different production types.”

Comment 6: For section 3.2, it would be helpful to know the exact amino acid length of the analyzed HA fragments and whether they were sequenced from the start. Does it include the full-length HA1 and partial HA2? Table 2 should be moved to supplementary material as it is very technical and already well-described in the text. Figures 3 and 4 are very similar, so please select only one to include in the main text,and move the other to supplementary material.

We agree with the Reviewer and thank you for bringing this to our attention.

We have provided clarification regarding the length of the analyzed HA fragments in section 2.4 of materiel and methods.

In addition, we made the following changes:

i)                We moved Table 2 to the supplementary material, as it contains technical details that have already been well-described in the main text,

ii)              We have carefully reviewed both and have decided to include Figure 4 in the main text, as it provides the most comprehensive and relevant information

iii)             We have moved Figure 3 to the supplementary materialas suggested by the reviewer.

Comment 7: In section 3.3, line 244 states that six strains were sequenced for HA1 and HA2. Please provide more details. Were these full-HA sequences? And were the others not sequenced? Line 253 seems to be a fragment, so please rephrase it. In Table 4, the footnote for a and b should be reversed.

We agree with the Reviewer.

The strains presented in table 4 were sequenced on the variable fragment for both HA1 and HA2 (between HA cleavage site) and presenting the main mutations noted on the AA. The strains where only HA2 were sequenced, only potential glycosylation sites who were investigated, as we hadn’t enough data to present the mutations on RBS.

To further clarify we added a paragraph in Results section 3-4 (line 243):

“The sequencing of the variable HA1 and HA2 regions were conducted for twelve strains out of the twenty-one sequenced. The strains where only HA2 were sequenced, only potential glycosylation sites who were investigated, as we hadn’t enough data to present the mutations on RBS. “

Reviewer 2 Report

Comments and Suggestions for Authors

Line 118-119: The sample numbers are odd. No explanation offered as to why those numbers were used.  Previous sentence states 282 samples were collected from 205 farms so the sampling plan was not uniform or the samples collected were not uniform. 

Line 165-170: I am not sure of the value of the percentages presented in this paragraph based upon the sampling plan. If farms were targeted to be suspect AI farms, why was the positivity rate so low? Some farms had multiple samples for testing giving them a greater chance for a detection to occur.

Figure 1. The distribution of farms was clearly in the north. No mention as to why. Are the poultry rearing regions only in the north? It would be helpful for international readers to understand why only certain regions were sampled. 

Line 188: West African strains "sch".........this should read "such"

Line 265: Why were only 7 regions included?

Line 274-275: So the positivity rate was lower than previous years. Explain the similarity between the two studies.....not just the fact that numerically, the positivity rate was lower.  Different time of year....different birds. I am not sure that difference can be explained by vaccination when this was not proven. What evidence can be offered to attribute the decrease in the positivity rate to the use of vaccines? 

Line 279: Are you implying that farms were identified for sampling because they had respiratory disease signs that turned out to be IBV and not AIV?  This drove up the number the farms sampled .....and found to be negative?  Were the samples that were found to be free of AIV tested for other pathogens?

Line 285: What are the central regions? According to the map, the central region (north to south) were not tested. 

Line 290: ....or perhaps the virus is evolving within the region/state? 

Line 293-294: Then should the countries work together to implement a vaccination program?

Line 297: A genetic diversity range was offered of 2-6%. This did not address tissue origin or bird type. Where viruses that were recovered from respiratory tissue different from gut? What about similarity or differences when looking at different species. 

Any comment on the genetic evolution of influenza when vaccination is not used compared to when it is used? How is this impacted by the vaccine employed, mainly as it pertains to HA identity? 

Comments on the Quality of English Language

I have no issues with the quality of the English language. 

Author Response

We thank the Reviewer for their valuable comments and the opportunity to improve our manuscript and resubmit to Viruses. We have carefully considered the comments and revised the manuscript accordingly. The following is a point by point response to the all the Reviewers’ questions and comments.

Comments 1: Line 118-119: The sample numbers are odd. No explanation offered as to why those numbers were used. Previous sentence states 282 samples were collected from 205 farms so the sampling plan was not uniform or the samples collected were not uniform. 

Thank you for your feedback and review of our sampling methodology.

In our sampling process, 282 samples were collected from 205 farms, this non-uniformity in sample sizes was due the high variability in the farms we surveyed, the suspected samples we received for H9N2 detection and the pooling methodology we established. A minimum of 10 samples were tested in each farm (depending on the farm size) then the samples were pooled to 1/3 pools. This non-uniform approach was designed to ensure that we captured a representative sample of the diverse conditions present in the study area.

To further clarify we added a paragraph in materiel and method section 2-1(line 115), it’s reads as follow:

“From our epidemiological surveys launched from September 2021 to March 2023 in seven regions of Morocco, we collected 1140 samples including (trachea and lungs) and oropharyngeal swabs which were pooled to 282 pool collected from205 farms (vaccinated and non-vaccinated) of different types of production (159 broiler, 24layer,10 breeder, and 12 turkey breeder). The samples were at least from 10 samples per farm and pooled subsequently to one or two pools for RT-PCR analysis. This pooling approach was designed to ensure that the farm's status (whether it is positive or negative) was represented adequately and accurately”.

Comments 2: Line 165-170: I am not sure of the value of the percentages presented in this paragraph based upon the sampling plan. If farms were targeted to be suspect AI farms, why was the positivity rate so low? Some farms had multiple samples for testing giving them a greater chance for a detection to occur.

Thank you for pointing this out, we agree with you.

Therefore, the percentages presented the positive farms (farms with at least one positive sample). The low positivity rate was explained later by the co-infections statues with other respiratory pathogens and vaccination statues. Furthermore, the aim of our study was to present the genetic diversity and the molecular mutations newly detected on the circulation strains in Morocco. We are working on larger epidemiological investigations, including a statistical representative sampling in the national territory will be hold and published in a separate paper.

Comments 3: Figure 1. The distribution of farms was clearly in the north. No mention as to why. Are the poultry rearing regions only in the north? It would be helpful for international readers to understand why only certain regions were sampled. 

Line 265: Why were only 7 regions included?

Line 290: …or perhaps the virus is evolving within the region/state? 

We agree with the Reviewer.

The selection of regions for sampling was primarily influenced by the presence or absence of respiratory signs and suspicion of H9N2 infection. We received samples and visited farms from the seven tested regions of Morocco, nonetheless due to the highest concentration of poultry farms in the north and west areas, the highest sampling and suspected farms were detected there. This is the reason why only seven regions were included.

Comment 4:Line 285: What are the central regions? According to the map, the central region (north to south) were not tested. 

The central region is replaced with north-west region. We have now added this precision in the manuscript on the discussion section (line 280)

“The focus on farms experiencing suspected health issues remains crucial as it allows for a deeper investigation into the specific challenges faced by these farms and can potentially aid in the development of more targeted interventions and control strategies. In addition, most of the investigated farms in our study were predominantly situated in Northwest regions (Rabat, Casablanca and nearby areas). This particularity was a direct result of the heightened poultry activity and growing industry in this geographical region. We acknowledge that the sampling distribution may not provide a diverse description of the entire national territory. Although, this focus allows us to gain insight into the areas where the virus is spreading most frequently.”

Comment 5: Line 274-275: So the positivity rate was lower than previous years. Explain the similarity between the two studies.....not just the fact that numerically, the positivity rate was lower.  Different time of year....different birds. I am not sure that difference can be explained by vaccination when this was not proven. What evidence can be offered to attribute the decrease in the positivity rate to the use of vaccines? 

We agree with the Reviewer. Thank you for your feedback, we want to provide some further clarifications.

The previous field studies in Morocco weren’t enough representative of the circulation of the virus in the national territory; a survey launched between 2016 and 2021 includes only 81 poultry farms (El mellouli et al, 2022) and a study launched in 2019 includes only 108 farms (Sikht et al, 2022). However, we were not able to gather sufficient information relative to the vaccines used and their composition to be able to conclude on this point.

To further clarify we modified a paragraph in discussion (line 275), it reads as follows:

“It is important to bring out that both studies included samples collected from suspected farms, presenting high mortality rates, respiratory signs, decrease in food consumption and a drop-in production. This decline in viropositivity rate could suggest several possibilities: changes in the prevalence of the virus, with a co-circulation of multiple respiratory pathogens with H9N2 virus especially infectious bronchitis virus [41], which cause the same respiratory clinical picture, improved biosecurity measures, or fluctuations in dis-ease dynamics within the poultry industry. Nonetheless, the vaccination programs implemented in poultry production sectors using local and non-local strains since the first introduction of the virus in Morocco in 2016 [16,42], may play an important role in this de-crease. Many studies revealed that adapted vaccination reduce the level and duration of virus shedding and increase the resistance of host infection [8,43]. Moreover, post-vaccination samples were found to indicate limited transmission of the virus in the field [33,46]. On the other hand, vaccination programs may result in faster antigenic drift of human and avian influenza virus [33,44]. Although, vaccination pressure and natural immune pressure due to endemic infections in unvaccinated flocks contribute to the evolution and genetic variation of HA gene segments [45].”

Comment 6: Line 279: Are you implying that farms were identified for sampling because they had respiratory disease signs that turned out to be IBV and not AIV?  This drove up the number the farms sampled ….and found to be negative?  Were the samples that were found to be free of AIV tested for other pathogens?

We agree with the Reviewer. The negative samples were subjected to further analysis for differential diagnostic with other respiratory pathogens especially co-infections with IBV. This will be presented in a separate study paper.

Comment 7: Line 293-294: Then should the countries work together to implement a vaccination program?

The Reviewer is correct.

Absolutely, H9N2 has become endemic in several African countries, especially in the neighboring North African countries. A vaccination program against AIV H9N2 must be implemented in poultry production sectors using different vaccine types, such as inactivated homologous vaccines and vectored vaccines. This approach started in Algeria, Tunisia, and Egypt in breeder, layers and turkey breeders but less extended in broilers(Jeevan et al. 2019; Larbi et al. 2022; Naguib et al. 2017)

To clarify this point we have added a paragraph in the discussion section; line (276), it’s reads as follow:

“Therefore, in the context of endemic LPAIV H9N2 in several African countries especially in the neighboring North Africa region, it is essential to extend vaccination programs to cover the entire poultry productions. This approach of vaccination in breeders, layers and turkey breeders has been launched in Algeria, Tunisia and Egypt but less extended in broilers [10,17,47]. For all that, updating of vaccine strains based on ongoing surveillance surveys and circulating genotypes is a crucial factor for epidemiological control of virus spreading and limiting H9N2 circulation.”

Comment 8: Line 297: A genetic diversity range was offered of 2-6%. This did not address tissue origin or bird type. Where viruses that were recovered from respiratory tissue different from gut? What about similarity or differences when looking at different species. 

We appreciate your thoughtful consideration of the genetic diversity analysis presented in our study. The genetic diversity range of 2-6% is a general overview and that a more detailed examination, which takes into account tissue origin and bird type, would provide a deeper understanding of the viral diversity. Our sampling was based only on respiratory tissues (trachea and lungs). We recognize the importance of investigating whether viruses recovered from respiratory tissues differ from those in the gut and examining any potential variations when comparing different avian species. These considerations are indeed critical for a more comprehensive analysis of the genetic diversity within the virus population. As a result, we will explore these factors in our further research to offer a more detailed and refined understanding of viral evolution and diversity.

Comment 9: Any comment on the genetic evolution of influenza when vaccination is not used compared to when it is used? How is this impacted by the vaccine employed, mainly as it pertains to HA identity? 

Thank you for your comments, to clarify this point we have added a paragraph in the discussion section; it’s reads as follow:

“Many studies revealed that adapted vaccination reduces the level and duration of virus shedding and increase the resistance of host infection [8,43]. Moreover, post-vaccination samples were found to indicate limited transmission of the virus in the field [33,46]. On the other hand, vaccination programs may result in faster antigenic drift of human and avian influenza virus [33,44]. Although, vaccination pressure and natural immune pressure due to endemic infections in unvaccinated flocks contribute to the evolution and genetic variation of HA gene segments [45]”